

# Chorein sensitive microtubule organization in tumor cells

Saad Alkahtani[1], Abdullah A. Alkahtane[1], Christos Stournaras[2] and Saud Alarifi[1]

[1] Department of Zoology, College of Science, King Saud University, Riyadh, Saudi Arabia
[2] Department of Biochemistry, University of Crete Medical School, Heraklion, Greece

## ABSTRACT

**Background.** The purpose of this study is to analyzed the involvement of chorein in microtubules organization of three types of malignant; rhabdomyosarcoma tumor cells (ZF), rhabdomyosarcoma cells (RH30), and rhabdomyosarcoma cells (RD). ZF are expressing high chorein levels. Previous studies revealed that chorein protein silencing in ZF tumor cells persuaded apoptotic response followed by cell death. In addition, in numerous malignant and non-malignant cells this protein regulates actin cytoskeleton structure and cellular signaling. However, the function of chorein protein in microtubular organization is yet to be established.

**Methods.** In a current research study, we analyzed the involvement of chorein in microtubules organization by using three types of malignant rhabdomyosarcoma cells. We have applied confocal laser-scanning microscopy to analyze microtubules structure and RT-PCR to examine cytoskeletal gene transcription.

**Results.** We report here that in rhabdomyosarcoma cells (RH30), chorein silencing induced disarrangement of microtubular network. This was documented by laser scanning microscopy and further quantified by FACS analysis. Interestingly and in agreement with previous reports, tubulin gene transcription in RH cells was unchanged upon silencing of chorein protein. Equally, confocal analysis showed minor disordered microtubules organization with evidently weakened staining in rhabdomyosarcoma cells (RD and ZF) after silencing of chorein protein.

**Conclusion.** These results disclose that chorein silencing induces considerable structural disorganization of tubulin network in RH30 human rhabdomyosarcoma tumor cells. Additional studies are now needed to establish the role of chorein in regulating cytoskeleton architecture in tumor cells.

Corresponding author
Saad Alkahtani,
salkahtani@ksu.edu.sa

## INTRODUCTION

Chorein proteins are large proteins that share structural features (*Segeletz et al., 2019*). Various cells and tissues express differentially chorein, which is the eventual protein product of Vacuolar Protein Sorting Protein 13A (VPS13A) (*Velayos-Baeza et al., 2004*; *Segeletz et al., 2019*). Lack of this protein causes/leads to the development of an autosomal and progressive neurodegenerative disease called chorea-acanthocytosis (ChAc) (*Dobson-Stone et al., 2004*; *Tomiyasu et al., 2011*; *Dobson-Stone et al., 2002*; *Ueno et al., 2001*). This disease is escorted by severe movement's disorders and variable erythrocyte acanthocytosis

(*Velayos Baeza et al., 1993*; *Saiki et al., 2007*; *Adjobo-Hermans, Cluitmans & Bosman, 2015*). Previous studies revealed that this protein is also linked with regulation of various functional cell responses (*Velayos-Baeza et al., 2004*; *Ueno et al., 2001*). These include exocytosis (*Lang et al., 2017*; *Pelzl et al., 2017*), secretion and aggregation (*Schmidt et al., 2013*), endothelial cell stiffness (*Alesutan et al., 2013*), cytoskeletal organization (*Honisch et al., 2015a*; *Shiokawa et al., 2013*) and cell survival (*Honisch et al., 2015b*). From these studies, it became evident that chorein may be implicated in various additional pathophysiological disorders besides neurodegenerative diseases.

Recent studies partially elucidated the molecular footing of the cytoskeletal alterations governed by chorein. Reportedly, chorein binds to phosphatidylinositol lipids (*Foller et al., 2012*). This interaction activates a downstream cascade involving phosphoinositide-3-kinase (PI3K)-p85-subunit, Ras-related C3 botulinum toxin substrate 1 (RAC1) activation followed by phosphorylation of p21-activated kinase 1 (PAK1) (*Foller et al., 2012*). These findings documented that chorein may be critical in regulating cytoskeletal architecture.

In contrast, alterations of cytoskeleton organization and signaling, as depicted formerly in various tumor cells (*Stournaras et al., 1996*; *Wang et al., 2022*), appears to be a crucial dedifferentiation step (*Araki et al., 2015*; *Stournaras et al., 2014*; *Kotula, 2012*; *Papakonstanti & Stournaras, 2008*). This interface may in turn govern key cellular functions of tumor cells. These include cell growth and apoptosis (*Grzanka, Grzanka & Orlikowska, 2003*; *Papadopoulou et al., 2009*), motility and invasiveness (*Nürnberg, Kollmannsperger & Grosse, 2014*; *Jiang, Enomoto & Takahashi, 2009*; *Yamazaki, Kurisu & Takenawa, 2005*), Epithelial to Mesenchymal Transition (EMT) (*Yilmaz & Christofori, 2009*) and ion channel activity (*Lang & Stournaras, 2014*; *Alevizopoulos et al., 2014*).

We have previously studied in detail chorein-actin cytoskeleton interactions, both in malignant and non-malignant cells (*Schmidt et al., 2013*; *Foller et al., 2012*). In addition, the function of chorein protein in modulating organization of microtubular and intermediate filaments has been investigated in non-malignant cells (*Honisch et al., 2015a*). Functional analysis revealed that chorein participates in the regulation of diverse cellular functions (*Saiki et al., 2007*) including exocytosis (*Pelzl et al., 2017*), cytoskeletal organization (*Honisch et al., 2015a*). In this study, we reported disordered microtubular cytoskeleton, and cytokeratins and desmin in fibroblasts extracted from patients with chorea-acanthocytosis (ChAc), when evaluated against fibroblasts from that of non-diseased donors (*Honisch et al., 2015a*). These data further supported a meaningful function of chorein protein as cytoskeletal regulators. However, up to now it is still elusive whether chorein may also be involved in the modulation of the microtubular architecture in tumor cells. In this research study, we thus addressed a potential part that chorein may play in regulating tubulin architecture in tumor cells. We report here that chorein protein silencing induced large deregulation of microtubules organization in diverse human rhabdomyosarcoma cancer cells, establishing a substantial role of chorein in tumors.

## MATERIALS & METHODS

### Cell line

Rhabdomyosarcomas represent soft tissue sarcomas that are common in childhood. In the present study we have used three rhabdomyosarcoma cell lines: (a) the ZF alveolar multifocal rhabdomyosarcoma cell line (established from Dr. Sabine Schleicher at the Children's Hospital Tuebingen), (b) the RD embryonal rhabdomyosarcoma cell line (DSMZ, Braunschweig, Germany) and (c) the alveolar rhabdomyosarcoma cell line RH30 (DSMZ, Braunschweig, Germany) were allowed to grow in Dulbecco's Modified Eagle Medium (DMEM) comprising of 10% fetal bovine serum, 1% L-glutamine, and was complemented with 1% penicillin-streptomycin at 37 °C temperature and 5% $CO_2$.

### Chorein protein silencing

Briefly, cells ($1 \times 10^5$) were sowed in six-well plates for 24 h prior to the process of transfection. Afterwards, the transfection was performed with siRNA for VPS13A (chorein) (ID# s23342; Ambion, Darmstadt, Germany) or with a control (negative) siRNA (ID#4390843; Ambion) through polyamine-based Ambion™ siRNA transfection agent (Ambion) as per manufacturer's guide.

### RT-PCR

Cells were dished at $3 \times 10^5$ cells/ml and nurtured for 48 h before isolating RNA in DMEM comprising of 10% FBS and 1% penicillin-streptomycin. To determine $\alpha$-tubulin and β-tubulin transcription, total RNA was carefully extracted at 24 h, 48 h, and 72-hours post-transfection through TriFast™ (Peqlab, Erlangen, Germany).

### FACS measurement

Quantification of microtubule network abundance was established through FACS Calibur flow cytometry (BD Biosciences, San Jose, CA, USA) exactly as described before (*Yu et al., 2016*). In brief, untreated and chorein-silenced RH30 rhabdomyosarcoma cells were removed, washed twice with phosphate buffered saline (PBS) solution, followed by fixation with 4% paraformaldehyde at room temperature for 20 min. After washing with 3% BSA in PBS, RH30 rhabdomyosarcoma cells were nurtured for another one hour with anti-tubulin primary antibody (Cell Signaling Technology, Danvers, MA, USA). After that staining with secondary FITC goat anti-rabbit antibody (1:500; Invitrogen, Waltham, MA, USA) was performed for 30 min at 37 °C. Finally, samples were assessed by the FACS flow cytometer.

### Confocal laser scanning microscopy

Concisely, cells stayed in PBS at $5 \times 10^7$ cells/ml. Of which, 10 µl of sample was taken on glass slide and dried for 30 min before fixation with methanol. After repeated PBS washing for 10 min, the sample was utilized. To look for microtubular staining, cells were fixated with 4% PFA at room temperature for 15 min. Another dual wash was performed with PBS. The slides were then incubated with blocking buffer (1xPBS/5% normal goat serum/0.3% Triton™ X-100) for 60 min at room temperature. Next, the samples were subjected to $\alpha$-tubulin antibody overnight at temperature 4 °C (1:50; Cell Signaling). The samples were then washed three times with PBS and incubated for 90 min with Goat

anti-Rabbit IgG (FITC) antibody (1:500; Invitrogen). Next, nuclei were dyed for 10 min using DRAQ-5 dye (Biostatus). Mounting was executed through ProLong® Gold Antifade reagent (Invitrogen). Lastly, Zeiss LSM 5 EXCITER was used for confocal microscopy and images were examined using the same instrument-embedded software.

### Statistical analysis

Research data are presented as arithmetic means $\pm$ SD $\pm$ SEM. Unpaired $t$-test was used to determine the statistical significance between two independent groups. $p < 0.05$ was regarded as a difference between groups that is statistically significant.

## RESULTS

We have analyzed microtubules architecture in different rhabdomyosarcoma cell types. In ZF cells, analysis of morphological changes using confocal laser microscopy failed to establish a clear reorganization of microtubular structures (Fig. 1), although diminished intensities in fluorescence staining of microtubules were apparent in chorein-silenced cells. Similar observations became evident in RD cells (Fig. 2).

In contrast, confocal laser microscopy revealed a clear microtubules reorganization in chorein-silenced RH30 rhabdomyosarcoma cells in contrast to untreated RH30 cells (Fig. 3). In order to further analyze this interesting finding, we performed quantification analysis of engagement of microtubules by using FACS based tubulin-tracker fluorescence measurements. As depicted in Fig. 4, a sizable decrease of tubulin based microtubular fluorescence staining became evident upon chorein silencing (Fig. 4), fully supporting the morphological observations (Fig. 3). This finding implies clearly depolymerization of microtubules upon silencing of chorein in RH30 rhabdomyosarcoma cells.

Interestingly, the morphological alterations in microtubular structures in chorein-silenced RH30 rhabdomyosarcoma cells were not paralleled by transcriptional control of tubulin gene. Indeed, as displayed in (Fig. 5), both alpha- and beta- tubulin RNA transcripts were not statistically different in wild type and chorein-silenced RH30 cells, conforming with previously reported findings. In this report, tubulin gene transcription in chorein-deficient human fibroblasts retrieved from ChAc patients was similar to fibroblast from non-diseased donors (*Honisch et al., 2015a*).

## DISCUSSION

Genetic alterations like mutation in chorine gene mostly lead to a rare neurodegenerative disease (*Segeletz et al., 2019*). In the current research, we are the first one to report that decrease in chorein protein, by silencing, downregulates the microtubular organization in tumor cells. This became evident in human rhabdomyosarcoma tumor cells with poorly differentiation. Those cells have previously been shown to possess high chorein expression (*Honisch et al., 2015b*). The most abundant effect was witnessed in the RH30 rhabdomyosarcoma cell line. Here, confocal laser scanning microscopy documented clearly the breakdown of microtubule organization upon chorein silencing (Fig. 3). Quantitation analysis by specific FACS based tubulin-tracker fluorescence measurement

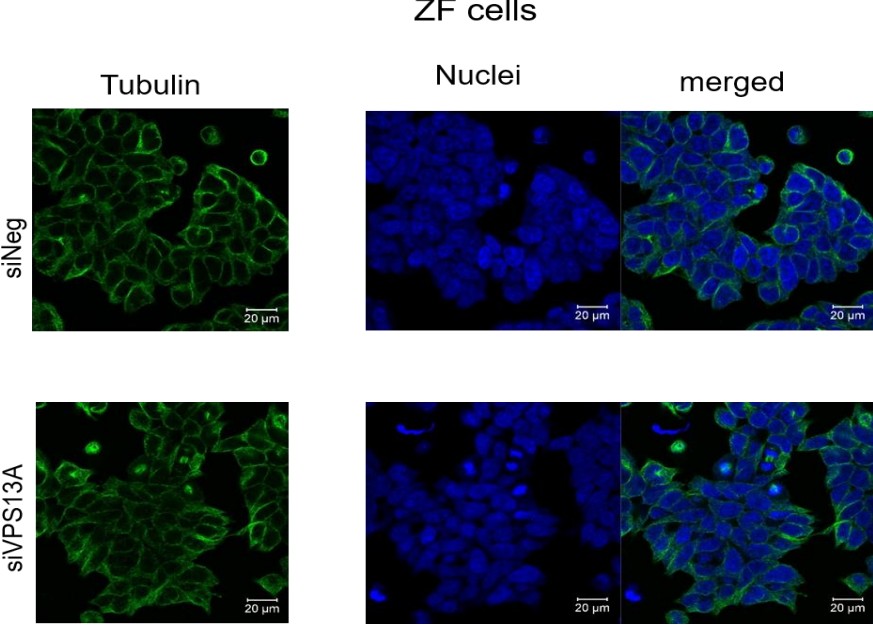

**Figure 1** Confocal laser scanning microscopy of α-tubulin and nuclei in chorein-silenced (siVPS13A) and control (siNeg) ZF rhabdomyosarcoma tumor cells. Scale bars represent 20 μm.

(Fig. 4) fully supported this effect in RH30 cells. The observed depolymerization of microtubules was however, much less prominent in ZF- and RD- rhabdomyosarcoma cells, although moderate diminished fluorescence staining was still observed by confocal microscopic analysis in both cell lines (Figs. 1 and 2). However, these moderate effects could not be supported by FACS based tubulin-tracker fluorescence analysis (data not shown). Since previous studies established differential expression levels of chorein in the three rhabdomyosarcoma cell lines (*Honisch et al., 2015b*), our present observations may pointing out a possible correlation between chorein expression levels and interaction to the microtubular network. Conversely, previous studies established that reorganization of cytoskeletal structures including actin microfilaments and modulation of cytoskeletal gene signaling in tumor cells may regulate pivotal cellular functions including cell survival and motility (*Yamazaki, Kurisu & Takenawa, 2005*; *Yilmaz & Christofori, 2009*; *Jaya, 2020*; *Ong et al., 2020*; *Li et al., 2022*). Accordingly, our study indicates that the microtubule reorganization might contribute to the reported regulation of apoptosis in rhabdomyosarcoma cells upon chorein silencing, as described before (*Honisch et al., 2015b*). Further studies are now necessary to confirm this interesting finding in additional tumor cell lines, expressing high levels of chorein.

Another interesting finding of the present work was the lack of transcriptional regulation of the tubulin gene upon chorein silencing. This result indicates that chorein downregulation modulates the polymerization equilibrium of microtubules rather than the protein gene expression.

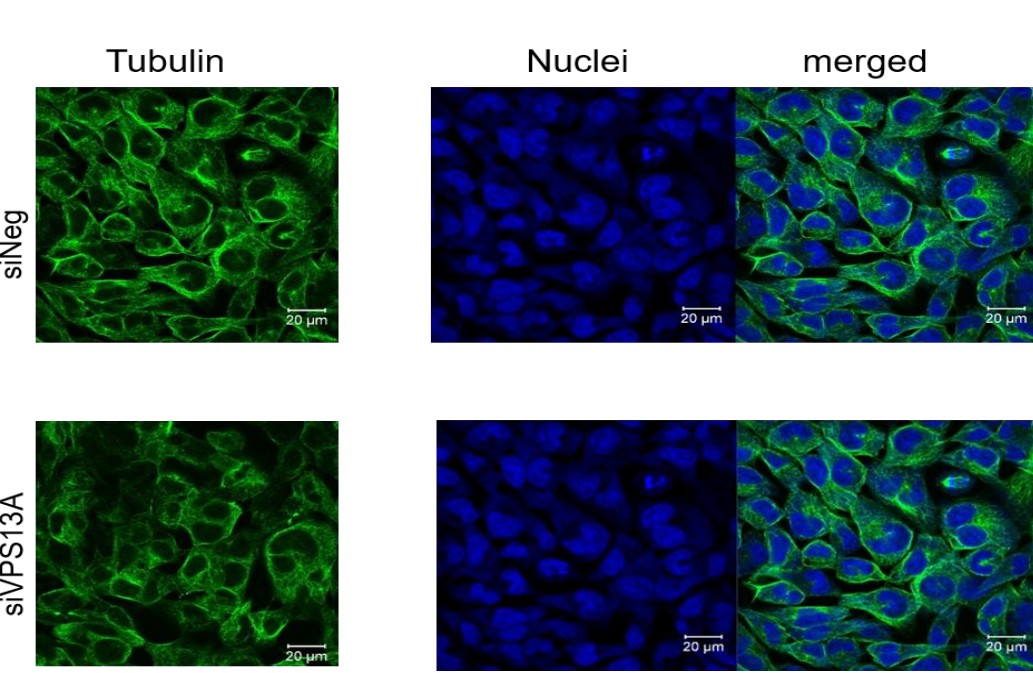

**Figure 2** Confocal laser scanning microscopy of α-tubulin and nuclei in chorein-silenced (siVPS13A) and control (siNeg) RD rhabdomyosarcoma tumor cells. Scale bars represent 20 μm.

This observation agrees with earlier research reports from human fibroblasts. Indeed, in these experiments tubulin gene transcription was not altered in fibroblast extracted from chorein protein deficient ChAc patients in comparison with the tubulin gene expression patterns in fibroblasts from non-diseased controls (*Honisch et al., 2015a*).

Our present findings showing a powerful regulation of tubulin-based cytoskeleton in rhabdomyosarcoma cells are associated with the recently reported interaction of chorein with actin cytoskeleton dynamics in the same cells (*Yu et al., 2016*). These findings imply a critical function of chorein protein in governing actin- and tubulin-base cytoskeletal architecture in tumor cells.

Besides tubulin and actin, chorein interacts as well with additional cytoskeletal proteins in erythrocytes or human fibroblasts. Indeed, a recent study on ChAc patients reported intracellular localization of both β-adducin and β-actin with chorein in human embryonic kidney 293 (HEK293) cells and erythrocytes expressing high levels of chorein protein. Since adducin along with actin regulate the function of synapsis, it was concluded that this co-localization might contribute to ChAc pathophysiology (*Shiokawa et al., 2013*). Another study reported that chorein may also interact with the intermediate filament structures of desmin and cytokeratins (*Honisch et al., 2015a*). Indeed, it was previously reported that actin microfilaments are depolymerized in erythrocytes, fibroblasts and blood platelets isolated from chorea-acanthocytosis (ChAc) patients (*Foller et al., 2012*; *Pelzl et al., 2017*).

**Figure 3** Confocal laser scanning microscopy of α-tubulin and nuclei in chorein-silenced (siVPS13A) and control (siNeg) RH30 rhabdomyosarcoma tumor cells. Scale bars represent 20 μm.

Furthermore, experiments performed in preparations of human fibroblasts extracted from patients with ChAc showed weakened expression of cytokeratins and desmin gene, along with reorganization of staining of cytokeratin- and desmin-intermediate filaments (*Honisch et al., 2015a*). Taken together, these data strongly imply that chorein protein is a potent regulator of numerous cytoskeletal elements. However, it remains elusive whether chorein interactions with additional cytoskeletal elements are likewise relevant in rhabdomyosarcoma tumor cells.

## CONCLUSIONS

This work elucidates for the first time the interaction between chorein and tubulin cytoskeleton in human RH30 rhabdomyosarcoma cell line. The effect was mainly manifested by the observed structural reorganization of microtubules following depolymerization. There was no experimental evidence directing the control of gene transcription of tubulin by chorein. Our findings establish an essential function of chorein protein in governing cytoskeletal structure in poorly differentiated tumor cells. Further studies are now needed to establish the clinical potential of those observations.

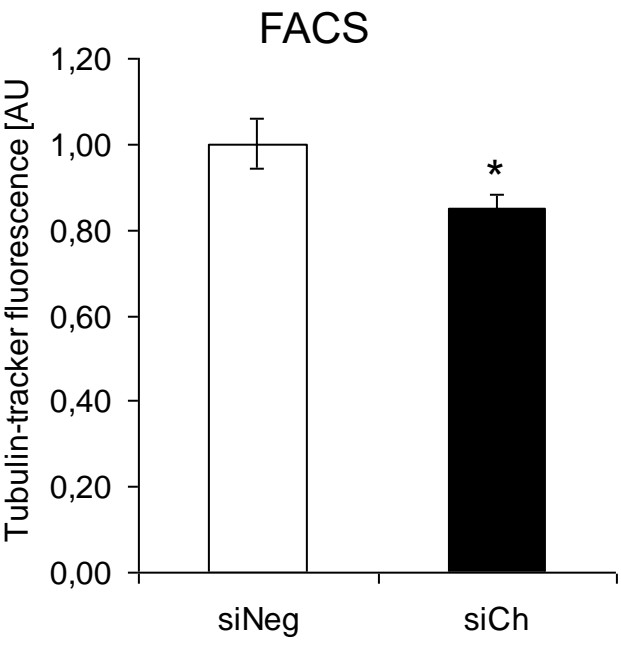

**Figure 4 Quantification analysis of the microtubular network by using flow cytometry-based tubulin-tracker fluorescence.** $^\star(p < 0.05)$ indicates statistical significance.

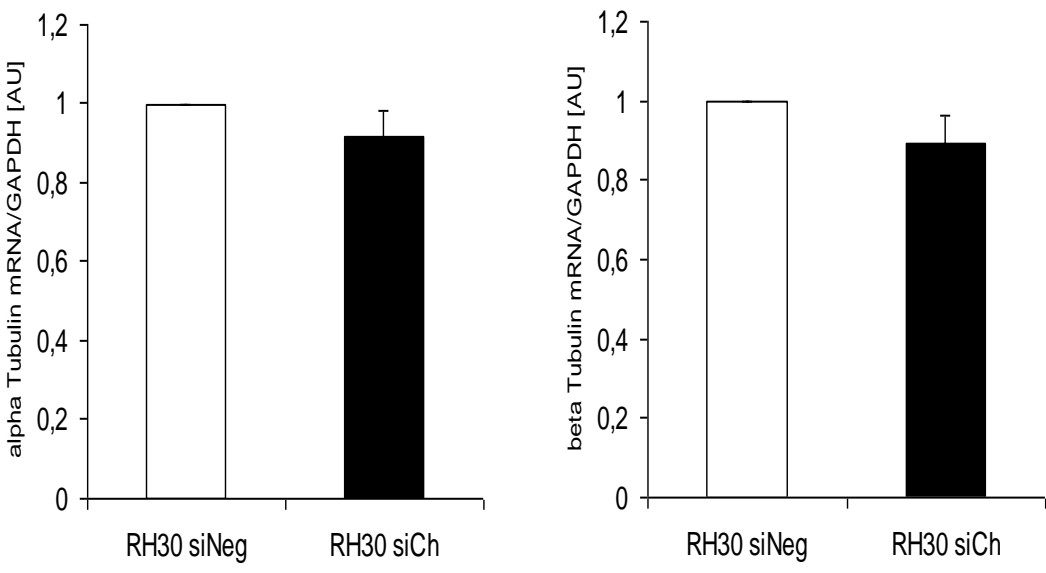

**Figure 5 Transcriptional expression of α-tubulin (left panel) and β-tubulin (right panel) in chorein-silenced (siVPS13A) and control (siNeg) RH30 rhabdomyosarcoma tumor cells.** Tubulin genes were measured by quantitative real time RT-PCR, with GAPDH as the internal control in cell lysates.

## ACKNOWLEDGEMENTS

Authors thank Dr. Sabina Honisch for the excellent support provided for this work.

### Funding

This work was fundedby the National Plan for Science, Technology and Innovation (MAARIFAH), King Abdul-Aziz City for Science and Technology, Kingdom of Saudi Arabia, grant number 14-MED-1893-02. The funders had no role in study design, data collection and analysis, decision to publish, or preparation of the manuscript.

### Grant Disclosures

The following grant information was disclosed by the authors:
the National Plan for Science, Technology and Innovation (MAARIFAH), King Abdul-Aziz City for Science and Technology, Kingdom of Saudi Arabia: 14-MED-1893-02.

### Competing Interests

The authors declare there are no competing interests.

### Author Contributions

- Saad Alkahtani conceived and designed the experiments, performed the experiments, authored or reviewed drafts of the article, and approved the final draft.
- Abdullah A. Alkahtane performed the experiments, analyzed the data, prepared figures and/or tables, and approved the final draft.
- Christos Stournaras performed the experiments, prepared figures and/or tables, authored or reviewed drafts of the article, and approved the final draft.
- Saud Alarifi performed the experiments, prepared figures and/or tables, and approved the final draft.

### Data Availability

    The raw data are available in the Supplemental File.

### Supplemental Information

Supplemental information for this article can be found online at http://dx.doi.org/10.7717/peerj.16074#supplemental-information.

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
