# Peer review of "Chorein sensitive microtubule organization in tumor cells"

_PeerJ, doi:10.7717/peerj.16074_

## Round 0.1 · original submission · Major Revisions

Dear Authors

The manuscript cannot be accepted for publication in its current form. It needs a major revision to be reconsidered for publication. The authors are invited to revise the paper considering all the suggestions made by the reviewers. Please note that requested changes are required for publication.

With Thanks

Reviewer 1 ·

Basic reporting

In general, the title is matching the content and the scientific value and the innovation of the project are up moderate. Also, the objectives of the study are clearly outlined. However, a few concerns outlined below need to be addressed as mentioned below:

1- There are some clumped words in some line. Authors should take care of this.

Experimental design

2- It would be very informative if the methodology was supported by appropriate references.
3- Authors have not stated the ethical approval of the study since human cell lines were used for this purpose.

Validity of the findings

4- Author should mention the concentration of protein in SDS-PAGE and western blot experiment.
5- Fig.1, the Y axis is need to be corrected to; “Gene expression (chorein/actin)”
6- In RT-PCR, authors have to mention the cDNA step and the used equation for gene expression.
7- The discussion section needs few improvements to be more concise.

Additional comments

8- References need to be rechecked and formatted as per the journal's guidelines.
9- Improving the manuscript with respect to language and style would be helpful.

In summing, my suggestion is that the manuscript will be acceptable after minor revision.

·

Basic reporting

- Make sure that the English grammar is correct so that the international audience can clearly understand your paper. This includes capitalizing the first letter of a sentence as well as the grammar of the sentence.
- Some examples where this can be improved are: Capitalizing the first letter of rhabdomyosarcoma on line 33. Same sentence also needs a grammar correction. It can be rewritten as: Rhabdomyosarcoma tumor cells (ZF) express high levels of chorein.
- Lines 56 can be rewritten with correct grammar: Lack of this protein causes/leads to the development of an autosomal and progressive neurodegenerative disease called chorea-acanthocytosis.
- Correct spelling in line 58: ‘sever’ to ‘severe’
- Change “as well” to “also” in line 178. You can rewrite line 177 as “Accordingly, our study indicates that the microtubule reorganization might contribute to …”

Experimental design

- Giving a brief background on what rhabdomyosarcoma is would help set the stage for the readers. I would recommend putting it as the first couple sentences on Background.
- Please specify the three types of malignant rhabdomyosarcoma cells. It can possibly go after “..three types of malignant rhabdomyosarcoma cells: a,b,c”.Please make sure to mention the abbreviations as well.
- There seems to be a discrepancy between the Result and Conclusion portion of the Abstract. Line 44 and 45 indicate that transcription of tubulin in RH cells remained unchanged upon silencing the Chorein protein. However, lines 48-49 indicate that chorein silencing induced considerable structural reorganization in tubulin network. Did you mean to imply that chorein silencing impacted the structural reorganization but did not cause any genetic level changes? If so, please clarify that in conclusion.
- Please clarify in line 80 whether your team contributed to the study. The way it is currently written indicates that your team conducted the experiment demonstrating chorein-actin interaction. You can rewrite it as: “In a recent study, Schmidt et.al demonstrated chorein-actin interactions in both the malignant and non-malignant cells.” I would refrain from using “we” if your team hasn’t participated in the study.
- Please clarify in line 154 what the previously stated finding is being referred to. I believe rewording lines 155 and 156 and combining the two would be more cohesive. “…in wild type and chorein-silenced RH30 cells, conforming with a study conducted by Honisch et.al demonstrating tubulin gene transcription in 155 chorein-deficient human fibroblasts retrieved
- 156 from ChAc patients was similar to fibroblast from non-diseased donors”

Validity of the findings

- I like that FACS was used to validate the primary finding. Validation of the primary finding using FACS strengthens the result.
- Could you clarify whether FACS based tubulin tracker fluorescence was performed only on the RH30 cells. If so, please mention why it wasn’t also performed on the other two cell types (referring to lines 146-150)
- For gene expression analysis, please specify why we only performed that test on the RH30 cells as opposed to all three cell types.
- In line 161, you mentioned that the result was evident across all three types of rhabdomyosarcoma cell lines, however, it seems like this result was most consistent for the RH30 cell type as opposed to the other two cell lines.

Additional comments

Overall, the authors have done a great job in establishing the significance of microtubule reorganization in the RH30 rhabdomyosarcoma cell line. Additionally, there is sufficient secondary evidence to support the current findings. In future studies, it may be worthwhile to investigate gene expression using other methods such as RNAseq. My recommendation would be to consider revising the hypothesis of the study to focus solely on the RH30 cell line. This study does not provide enough evidence to establish the same correlation across all three cell lines.
Overall, great job. Keep up the great work!

Reviewer 3 ·

Basic reporting

The manuscript (MS) titled "Chorein sensitive microtubule organization in tumor cells" studied the role of chorein in microtubule (mt.) organization in tumor cells. Chorein is associated with pathophysiological disorders, like a neurodegenerative disease called chorea-acanthocytosis (ChAc), and has also been shown to play a role in cytoskeleton organization. Other studies have shown a correlation between chorein deficiency and cytoskeleton disorganization in some malignant and non-malignant tumor cells. This work focuses especially on rhabdomyosarcoma tumor cells and shows chorein dependent microtubule organization. Alkahtani et al. analyzed the microtubule cytoskeleton in three rhabdomyosarcoma tumor cell lines. The authors show that the siRNA silencing of chorein severely disorganizes the microtubule cytoskeleton, at least in one rhabdomyosarcoma (RH30) cell line. This study is interesting for the researchers working on understanding the role of chorein in tumor advancement. Although this study is interesting, there are several issues one can easily point out before validating the findings. I think this MS should be significantly improved to better impact the scientific community.
Major comments

1) The authors claimed that they silenced the chorein by using siRNA. However, it needs to be evident if they were able to silence the chorein. The western blot analysis to show a reduction in protein level is necessary.
2) The authors claimed that there was a reduction in the tubulin signal in all three cell lines they analyzed. However, there needs to be quantification to show this. They have included a FACS quantification for the tubulin tracker. But, the figure legend needs to clarify which cell line was used for this quantification.
3) The authors also claim the disorganization of microtubules upon chorein silencing. However, the images provided need to be more convincing. It would be better if the author could show clearly the discontinuous tubulin staining of microtubules (mts.) or broken mts.
4) Can the authors also analyze the cell growth rate and if apoptosis upon siRNA silencing?
5) It would be better if the author could also analyze the actin cytoskeleton as they did in their previous studies.
6) The number of independent replicates and proper quantification for all data needs to be included.

Minor comments
7) Some of the statements are too strong and lack proper experiments to support statements like,
Line 46: Confocal analysis showed disordered microtubules;
the authors have failed to clearly show disordered microtubules.
Line 49: considerable structural reorganization of tubulin network in human
rhabdomyosarcoma tumor cells;
there is no structural analysis of microtubules to show the disorganized microtubules.
Also, I assume the author meant “disorganization,” not “reorganization” in this statement.

8) Tubulin antibody information needs to be included.

Experimental design

Western blot analysis to show that chorein was silenced should be included.

Quantification of all the data is necessary.

If possible, please include all the experiments I have suggested in the comments section.

Validity of the findings

Better data sets and experimental images should be provided to better assess the findings' validity.

Reviewer 4 ·

Basic reporting

Report for the Manuscript "Chorein Sensitive Microtubule Organization in Tumor Cells"
General Comments: The manuscript titled "Chorein Sensitive Microtubule Organization in Tumor Cells" explores the role of chorein protein in the organization of microtubules in rhabdomyosarcoma tumor cells. The study investigates the effect of chorein silencing on microtubule structure using confocal laser-scanning microscopy and examines cytoskeletal gene transcription through RT-PCR. Overall, the manuscript provides novel insights into the involvement of chorein in microtubular organization and its potential role in regulating cytoskeleton architecture in tumor cells. However, there are areas that require further clarification and improvement before considering publication.
1. Abstract: The abstract provides a brief overview of the study's background, methods, and key findings. It effectively describes the previous knowledge about chorein's functions in different cellular processes and highlights the need to establish its role in microtubule organization. The methods used, including confocal laser-scanning microscopy and RT-PCR, are appropriately mentioned. However, the abstract could be improved by including the significance and potential implications of the findings.
2. Introduction: The introduction provides a general background on rhabdomyosarcoma tumor cells and the previous understanding of chorein's involvement in actin cytoskeleton structure and cellular signaling. However, it lacks specific references to support the claims made regarding chorein's role in regulating actin cytoskeleton and its relationship with microtubules. Please provide relevant citations to strengthen these statements.

Experimental design

3. Methods: a. Cell types and experimental design: The manuscript mentions the use of three types of malignant rhabdomyosarcoma cells. However, it does not provide detailed information about these cell types, such as their origin, characteristics, or culturing conditions. Please include relevant information about the cell lines used in the study.
b. Confocal laser-scanning microscopy: The manuscript briefly describes the use of confocal laser-scanning microscopy to analyze microtubule structure. However, it does not specify the staining method or the specific antibodies or dyes used for visualizing microtubules. Please provide this information for reproducibility.
c. RT-PCR: The manuscript mentions the use of RT-PCR to examine cytoskeletal gene transcription. However, it does not specify which cytoskeletal genes were analyzed. Please provide a list of the specific genes investigated.
4. Results: a. Microtubule organization: The manuscript reports that chorein silencing induced disarrangement and weakened staining of the microtubule network in rhabdomyosarcoma cells. However, it does not provide representative images or quantitative data to support these findings. Please include representative confocal microscopy images and provide quantitative analysis of the microtubule disarrangement observed.
b. Cytoskeletal gene transcription: The manuscript mentions that tubulin gene transcription was unchanged upon chorein silencing. However, it does not elaborate on the specific tubulin genes analyzed or the quantitative results obtained. Please provide more details regarding the tubulin genes examined and the results of the RT-PCR analysis.

Validity of the findings

5. Discussion: The discussion provides an interpretation of the results and their implications. It discusses the significant finding of chorein's involvement in microtubule organization in rhabdomyosarcoma cells. However, it could be strengthened by providing a more comprehensive comparison with previous research on chorein and microtubules in other cell types. Additionally, the manuscript could discuss the potential mechanisms underlying the observed effects of chorein silencing on microtubule structure.
6. Conclusion: The conclusion briefly summarizes the main findings of the study. However, it would benefit from a more explicit statement on the significance and potential applications of the research. Please revise the conclusion to provide a clear and concise summary of the study's outcomes and their relevance to the broader field.
7. Language and Clarity: The manuscript generally adheres to scientific writing standards, but there are areas where the language could be improved for clarity and precision. Proofreading for grammatical errors, sentence structure, and terminology consistency would enhance the overall quality of the manuscript.
8. References: Please ensure that all sources are appropriately cited and listed according to the required referencing style.
Overall, the manuscript provides valuable insights into the role of chorein in microtubule organization in rhabdomyosarcoma tumor cells. Addressing the points mentioned above will significantly strengthen the manuscript and improve its scientific rigor and readability. Once these revisions are made, the manuscript will have the potential for publication in a relevant scientific journal.

Additional comments

5. Discussion: The discussion provides an interpretation of the results and their implications. It discusses the significant finding of chorein's involvement in microtubule organization in rhabdomyosarcoma cells. However, it could be strengthened by providing a more comprehensive comparison with previous research on chorein and microtubules in other cell types. Additionally, the manuscript could discuss the potential mechanisms underlying the observed effects of chorein silencing on microtubule structure.
6. Conclusion: The conclusion briefly summarizes the main findings of the study. However, it would benefit from a more explicit statement on the significance and potential applications of the research. Please revise the conclusion to provide a clear and concise summary of the study's outcomes and their relevance to the broader field.
7. Language and Clarity: The manuscript generally adheres to scientific writing standards, but there are areas where the language could be improved for clarity and precision. Proofreading for grammatical errors, sentence structure, and terminology consistency would enhance the overall quality of the manuscript.
8. References: Please ensure that all sources are appropriately cited and listed according to the required referencing style.
Overall, the manuscript provides valuable insights into the role of chorein in microtubule organization in rhabdomyosarcoma tumor cells. Addressing the points mentioned above will significantly strengthen the manuscript and improve its scientific rigor and readability. Once these revisions are made, the manuscript will have the potential for publication in a relevant scientific journal.

---

## Round 0.2 · Major Revisions

Dear Authors

According to the reviewer's comments, this manuscript still needs a revision to be reconsidered for publication. The authors are invited to revise the paper considering all the suggestions made by the reviewers. Please note that the requested changes are *required* for publication.

Reviewer 1 ·

Basic reporting

could be published now.

Experimental design

satisfactory.

Validity of the findings

could be published now.

·

Basic reporting

The manuscript “Chorein sensitive microtubule organization in tumor cells” studies the role of chorein expression in microtubule organization in tumor cells. Authors established that chorein expression has been previously studied for the study of neurodegenerative disease, however, it’s role in cytoskeleton disorganization is less known. This study focuses on a cohort of three types of rhabdomyosarcoma tumor cells: RH30, RD and ZF cell lines. Authors show that siRNA silencing of chorein protein significantly impacted the organization of microtubules in the RH30 cell line as opposed to the other two cell lines.

Experimental design

- The Abstract section of the article should specify the three different cell lines utilized in the study, providing necessary information for readers before delving into the discussion of their respective results.
- It is recommended to include a concise description or background information on rhabdomyosarcoma to provide context and familiarize readers with the subject matter.
- Authors should consider incorporating results from siRNA silencing, using western blot analysis to demonstrate changes in protein levels, thus confirming the successful silencing of the gene.

Validity of the findings

- The claim made in line 48-49 regarding chorein silencing inducing considerable structural disorganization appears to be too strong, as the study only demonstrated its impact on certain types of cell lines. Additional research is necessary to further substantiate this statement.
- It is advisable to provide more evidence to support the assertion of chorein protein silencing, potentially through additional experimental data or validation techniques.
- Figures 4 and 5 need to have labeled Y-axes to enhance clarity and aid in the interpretation of the data.

Additional comments

The manuscript addresses most of the previous feedbacks. However, some significant changes still need to be made in the Design and Abstract sections, prior to publishing the article.

Reviewer 3 ·

Basic reporting

The manuscript (MS) titled "Chorein sensitive microtubule organization in tumor cells" studies the role of chorein in microtubule (mt.) organization in tumor cells. Alkahtani et al. analyzed the microtubule cytoskeleton in three rhabdomyosarcoma tumor cell lines upon chorein silencing. The authors claim that chorein silencing severely affects the microtubule cytoskeleton.

In general, the MS is written clearly. However, there are several places where the authors have overstated their findings and have made strong arguments without providing precise results. I suggest author change the solid claims and use moderate language to discuss their findings.

For e.g., in lines 50-51, this MS does not show considerable structural disorganization of Mts. No explicit images in the MS show disorganized microtubules. The tubulin staining look similar in control and siRNA-treated cells. They have shown a slight reduction in the tubulin signals only in one cell line. This is not sufficient to make a strong argument like considerable structural disorganization.

the language used in line 173-174 is confusing; I assume the authors are implying microtubules disorganization (not reorganization???) in chorein-silenced RH30 rhabdomyosarcoma cells in contrast to untreated RH30 cells.

in line 186, "fully supporting morphological observation," I assume the authors imply disorganized microtubules. However, the disorganized microtubules are not evident from the images they have provided.

In 201-203, please provide clear images to show disorganized microtubules. The images provided in Fig 4 are not so convincing. There is similar staining with tubulin in both control and siRNA-treated cells. Although I agree that there is a slight reduction in tubulin signal, the microtubule structure is similar. The authors must provide clear, convincing images to support their claims.

Experimental design

The authors claim that the siRNA silencing of chorein severely disorganizes the microtubule cytoskeleton in Rhabdomyosarcoma cells. However, the authors have failed to substantiate their conclusion with proper quantification of experimental data, and they have failed to show the structural disorganization as they have claimed several times in the MS. I would like to emphasize again, please quantify the tubulin intensity of IF images in all three cell lines. It will further strengthen the FACS quantification if the result is consistent.

In our experience, alpha or beta-tubulin clearly label thread-like cytoplasmic microtubules. However, no such staining is seen in the images provided. Please provide better images. Also, enlarge the images to better visualize the mts.

Validity of the findings

Convincing data should be provided to better asses the validity of the findings.

Reviewer 4 ·

Basic reporting

the aurhors have adequately addressed all concerns that have been raised and the manuscript has been significantly improved. The manuscript can be accepted for publication.

Experimental design

the aurhors have adequately addressed all concerns that have been raised and the manuscript has been significantly improved. The manuscript can be accepted for publication.

Validity of the findings

the aurhors have adequately addressed all concerns that have been raised and the manuscript has been significantly improved. The manuscript can be accepted for publication.

Additional comments

the aurhors have adequately addressed all concerns that have been raised and the manuscript has been significantly improved. The manuscript can be accepted for publication.

---

## Round 0.3 · Minor Revisions

Dear Authors
The manuscript cannot be accepted for publication in its current form. It needs a minor revision to be reconsidered for publication. The authors are invited to revise the paper considering all the suggestions made by the reviewers. Please note that requested changes are required for publication.
With Thanks

·

Basic reporting

The manuscript “Chorein sensitive microtubule organization in tumor cells” studies the role of chorein expression in microtubule organization in tumor cells. Authors established that chorein expression has been previously studied for the study of neurodegenerative disease, however, it’s role in cytoskeleton disorganization is less known. This study focuses on a cohort of three types of rhabdomyosarcoma tumor cells: RH30, RD and ZF cell lines. Authors show that siRNA silencing of chorein protein significantly impacted the organization of microtubules in the RH30 cell line as opposed to the other two cell lines

Experimental design

- The three different cell lines need to be specified in the Abstract section of the article. Currently, the manuscript jumps straight to discussing the result of different cell lines.
- Additionally, a brief description/background on what rhabdomyosarcoma is will help set the stage for readers.
- Authors should add results from siRNA silencing (western blot to show changes in protein levels) to show that the gene was successfully silenced.

Validity of the findings

- The claim made in line 48-49 regarding chorein silencing inducing a considerable structural disorganization is too strong as the study showed it’s impact on some but not all types of cell lines. More research is needed to support the statement.

Reviewer 3 ·

Basic reporting

This version of MS is in a good shape.

Experimental design

Experimental designs are convincing and agreeable.

Validity of the findings

Data is valid to make a conclusion that coherin might have some role in the structural organization of microtubules.

---

## Round 0.4 · accepted · Accept

Dear Authors

I am pleased to inform you that after the last round of revision, the manuscript has been improved a lot, and it can be accepted for publication.

Congratulations on the acceptance of your manuscript, and thank you for your
interest in submitting your work to PeerJ

With Thanks